# Optimization of Aeration Rate—Low Cost but High Efficiency Operation of Aniline-Degrading Bioaugmentation Reactor

**Jianyang Song [1,2,3,*], Chunyan Wang [1,4], Haojin Peng [5], Qian Zhang [6], Yao Li [6], Hua Wei [6] and Hongyu Wang [3]**

1   Henan Key Laboratory of Industrial Microbial Resources and Fermentation Technology, Nanyang Institute of Technology, Nanyang 473004, China
2   School of Civil Engineering, Nanyang Institute of Technology, Nanyang 473004, China
3   School of Civil Engineering, Wuhan University, Wuhan 430072, China
4   School of Biological and Chemical Engineering, Nanyang Institute of Technology, Nanyang 473004, China
5   State Key Laboratory of Pollution Control and Resources Reuse, School of Environmental Science and Engineering, Tongji University, Shanghai 200092, China
6   School of Civil Engineering and Architecture, Wuhan University of Technology, Wuhan 430070, China
*   Correspondence: songjianyang66@163.com

**Abstract:** In this work, two sequencing batch reactors (R0 and R1) were built for treating 600 mg·L$^{-1}$ aniline-containing wastewater. R1 was a bioaugmentation system with the addition of highly efficient aniline-degrading strain AD4 (*Delftia* sp.), while R0 served as a control system. The effects of aeration rates on R1 and R0 were investigated in the range of 300 to 800 mL·min$^{-1}$. Results showed that the increase in aeration rates promoted the degradation of aniline in both R1 and R0. Under bioenhancement, the highest removal efficiency of aniline was stabilized above 99.7% in R1 while it was lower than 95.6% in R0. As for nitrogen removal, increasing aeration rates reduced the NH$_4^+$-N released from aniline degradation but caused the accumulation of NO$_3^-$-N and NO$_2^-$-N. R1 had a better total inorganic nitrogen removal efficiency than R0. The alpha biodiversity of both R1 and R0 reached the highest at 400 mL·min$^{-1}$ and decreased at higher aeration rates. However, R1 always kept higher biodiversity than R0. Furthermore, the abundance of various functional bacteria was also higher in R1. This study revealed the high efficiency of bioenhanced activated sludge for the treatment of refractory wastewater and illustrated the importance of aeration control from the angle of energy saving, which demonstrated the potential of biofortification to help energy conservation and emission reduction.

**Keywords:** aniline; bioaugmentation; aeration rate; high throughput sequencing; microbial community succession



## 1. Introduction

Aniline is a kind of refractory aromatic compound widely used in industrial manufacturing such as dyes, pesticides, rubber, herbicides and medicine, serving as a precursor or intermediate [1–3]. The high toxicity and carcinogenicity of aniline make it a huge threat to environmental safety and public health [4,5]. Except for this, the refractory nature of aniline in the environment and its features of volatility make it easy to spread and hard to be removed [6]. Proper treatments with high efficiency and low cost are needed before discharging aniline wastewater from wastewater treatment plants (WWTPs) to the environment [7]. Traditional treatments, such as adsorption, extraction and electrochemical treatments, are usually limited by their small scale, intensive energy demand and potential secondary pollution. There are also some studies using bioelectrochemical technologies to treat similar wastewater [8,9]. By contrast, biological processes are more promising and widely used in WWTPs because of their larger capacity and economic benefits [10,11].

However, aniline and its metabolites such as catechol were found to be toxic to microbes [12,13]. This makes this kind of wastewater much more intractable at a high

concentration of aniline. Bioaugmentation technologies, which are environmentally friendly and easy to operate, were usually used to help degrade refractory pollutants [14], so that occurred in the treatment of aniline wastewater. In recent years, many aniline-degrading bacteria have been isolated, identified and studied. For instance, a strain of the genus *Rhodococcus* was reported to completely degrade 1500 mg·L$^{-1}$ phenol or 800 mg·L$^{-1}$ aniline within 48 h [15]. Another kind of strain belonging to the genus *Ochrobactrum* was found to be resistant to 6500 mg·L$^{-1}$ aniline and could reach a more than 75% degradation rate in aniline concentration from 200 to 1600 mg·L$^{-1}$ [16]. These highly efficient aniline-degrading strains could be put into bioreactors to strengthen the removal performance of aniline. However, the analysis of functional flora at the phylum and genus levels was limited in the previous studies to the performance of aniline degradation biosystems [17]. The existing research results need to continue to make efforts on the microscopic effects of functional bacteria in bioaugmentation systems in discussing the effects of bioaugmentation on activated sludge [18].

From our earlier study, a highly effective aniline-degrading strain named AD4 (*Delftia* sp.) with an NCBI number (MK336721) was isolated and proven to greatly shorten the start-up period of the sequencing batch reactor (SBR) with a loading of 600 mg·L$^{-1}$ aniline in the influent [17]. To further promote this technology and for practical application consideration, it is necessary to optimize the operating conditions of the reactor. For the actual biological treatments, the aeration rate is an important operational parameter that determines the condition of the oxygen supply [19]. During the last few years, the aeration rate has been widely studied and found to be closely related to organic matter degradation and nitrogen removal [20,21]. Actually, aniline degradation is a process that depends on dissolved oxygen (DO), which is supplied by aeration. Furthermore, ammonia nitrogen, a byproduct of aniline degradation, is a common pollutant coexisting in aniline-containing wastewater [22]. It should also be considered for removal through ammonia-oxidizing bacteria, whose activities are influenced by the aeration rate too. This suggests the importance of investigating the effects of varying aeration rates on such an enhanced system with the bioaugmentation technology to degrade aniline.

In order to study the response of biosystem performance and microbial community changes under the aeration rate gradient, more detailed microbial sampling and analysis were designed in this study. The effects of the aeration rate gradient and bioaugmentation on the operation of the activated sludge system were evaluated from various aspects through the combination of macro performance and micro community changes. The main purposes of this work are listed as follows: (1) exploring the effects of varying aeration rates on the bioaugmentation and control systems treating a high concentration of aniline wastewater; (2) finding the optimal aeration rate in terms of the indexes of the effluent of aniline, $NH_4^+$-N, $NO_3^-$-N, $NO_2^-$-N and total inorganic nitrogen (TIN); (3) investigating the effects of the aeration rate on microbial community succession in terms of microbial diversity, structure and functional microorganisms. Theoretical guidance could be provided for the actual aniline wastewater treatment project, including aeration energy consumption and system performance management, so as to reduce the risk of aniline for ecological environment safety.

## 2. Materials and Methods

### 2.1. Reactors' Operation and Experimental Design

Two SBRs with a total volume of 1.5 L and a working volume of 1.0 L were established in the present work. The operation period of the SBR reactors was 8 h, which contained 0.5 h for influent, 6 h for aeration and stir, 1 h for sedimentation and 0.5 h for discharging effluent. The hydraulic retention time (HRT) was 16 h with a 50% water exchange ratio. The solids retention time (SRT) was about 10 d. Aeration pumps and glass rotameters were used to control the aeration rate in the experiment. The temperature was controlled at 28 ± 1 °C during the whole experiment. The seed sludge was collected from the aerobic

tank of a WWTP in Wuhan, Hubei Province, China. The mixed liquor suspended solids (MLSS) were measured periodically and maintained at $3000 \pm 500$ mg·L$^{-1}$.

The bacterial fluid of the AD4 strain was enriched until the value of OD$_{600}$ was 0.6~0.8. Then, a 5% volume fraction of AD4 bacterial fluid was added to the bioaugmentation system (R1). Another SBR reactor without adding bacterial fluid was served as a control group and named as R0. To study the effect of aeration rates on the aniline-degrading reactor, aeration conditions were set as an increase of 100 mL·min$^{-1}$ from 300 mL·min$^{-1}$ to 800 mL·min$^{-1}$. Both R1 and R0 were operated for 10 d continuously under each condition.

### 2.2. Synthetic Wastewater Composition

A synthetic medium in the influent was prepared with distilled water, and the main characteristics of the synthetic wastewater in the influent were: $NaH_2PO_4 \bullet 2H_2O$ 0.2600 g·L$^{-1}$, $Na_2HPO_4 \bullet 12H_2O$ 1.0079 g·L$^{-1}$, $(NH_4)_2SO_4$ 2.000 g·L$^{-1}$, $MgSO_4 \bullet 7H_2O$ 1.6367 g·L$^{-1}$, KCl 1.6000 g·L$^{-1}$, $Fe(NO_3)_3 \bullet 9H_2O$ 0.1069 g·L$^{-1}$ and aniline 0.6000 g·L$^{-1}$.

### 2.3. Chemical Analytical Methods

The dissolved oxygen (DO) and pH value were periodically measured in the six phases by a dissolved oxygen and pH monitor (HQ440d, HACH Water Quality Analytical Instrument Co., Ltd., Loveland, CO, USA) during the experiment. The analyses of aniline, $NH_4^+$-N, $NO_2^-$-N and $NO_3^-$-N were conducted according to the standard method [23]. All liquid samples were filtered through the 0.45 μm cellulose acetate filter first and measured in parallel.

### 2.4. Microbial Community Analysis

Sludge samples were named as C3, C4, C7, C8 and S3, S4, S7, S8 (C represents control group and S represents the bioaugmentation group) collected on the 10th, 20th, 50th and 60th day, respectively. All the sludge samples were stored and frozen at $-80$ °C until total genomic DNA extraction.

Total genomic DNA was extracted through the QIAamp-DNA kit. A nanodrop® ND-1000 spectrophotometer (Nanodrops® Technologies, Wilmington, DE, USA) was used to monitor the integrity, purity and concentration of the DNA. The primers for 16S rRNA amplification were selected as follows: 338F: 5′-ACTCCTACGGGAGGCAGC A-3′ and 806R: 5′-GCACTACHVGGGTWTCTAAT-3′. The Polymerase chain reaction was performed and the products of the eight samples were sent for further sequencing on the Illumine Miseq platform (Shanghai Majorbio Bio-pharm Technology Co., Ltd., Shanghai, China). The MOTHUR program was taken to carry out the operational taxonomic unit (OTU) with a setting value of 97% similarity.

## 3. Results and Discussion

### 3.1. Effects of Aeration Rates on Aniline Degradation

It can be clearly seen from Figure 1 that R1 showed amazing adaptability and aniline-degrading performance at phase I, and its degradation efficiency reached 97.9%. However, the aniline-degrading efficiency of R0 at phase I was only 79.9%. Although the aniline-degrading efficiency of R0 in phase I had an obvious improvement trend, the aniline concentration in the final effluent remained at about 100 mg·L$^{-1}$. The comparison between these two reactors undoubtedly reflected the powerful adjustment ability of biological reinforcement. The efficient aniline-degrading bacteria AD4 could rapidly capture aniline after being added to R1 at start-up, reducing the toxicity inhibition effect of aniline on activated sludge and accelerating the adaptability of the whole reactor to aniline wastewater. AD4 might be responsible for the diversion of aniline, which decreased the demand for DO supply from the activated sludge. A previous study also found that the aniline degradation efficiency of SBR supplemented with highly efficient aniline-degrading bacteria was much higher than that of the control reactor under the same aeration condition [24].

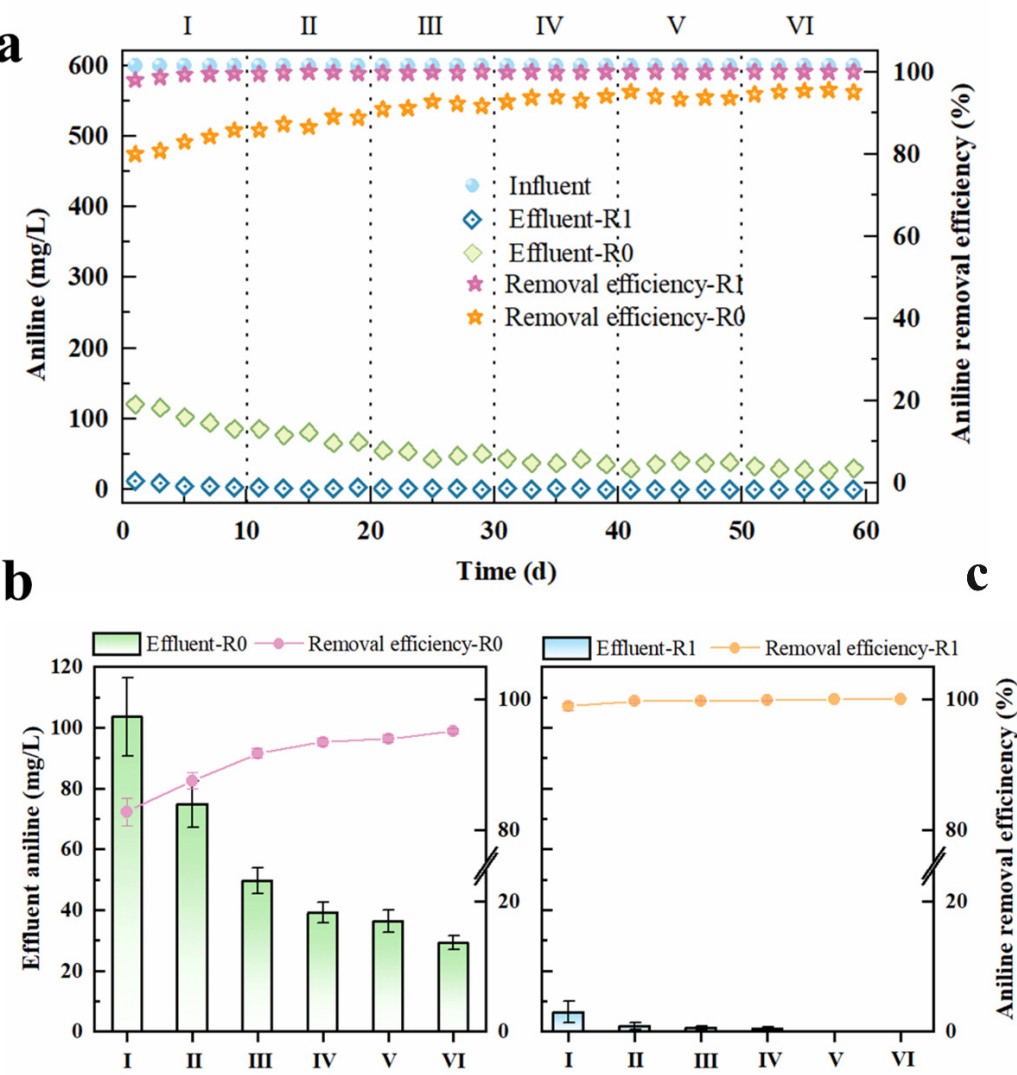

**Figure 1.** Removal performances of aniline in R0 and R1 (**a**); averages of effluent aniline concentrations and aniline removal efficiency of R0 (**b**) and R1 (**c**) from all days at each phase (Phase I to VI corresponding to 300 mL·min$^{-1}$ to 800 mL·min$^{-1}$ aeration rates, respectively).

As the aeration rate continued to rise, the aniline degradation efficiency of R1 increased from 98.08% to 100%. When the aeration rate reached 600 mL·min$^{-1}$ and 700 mL·min$^{-1}$, the aniline concentration could be around 1 mg·L$^{-1}$ or no longer be detected in the effluent and remained steady until the end of the operation. This also indicated that the aeration shear force caused by the aeration rate within 800 mg·L$^{-1}$ could not cause damage to sludge. Meanwhile, the increase in the aeration rate had a continuous promotion effect on the aniline-degrading performance of R0. From phase I–VI, the average aniline effluent concentration of R0 decreased from 103.58 ± 12.92 mg·L$^{-1}$ to 29.22 ± 2.31 mg·L$^{-1}$. This indicated that increasing the oxygen supply was an effective measure to enhance the aniline degradation in an activated sludge system. In the toxicity environment, on the one hand, the damaged microorganism needed to increase the synthesis of essential substances for survival, such as nucleic acid and protein, to repair cell structure and maintain physiological metabolism. On the other hand, activated sludge often stimulated the secretion of substances, such as extracellular polymer substances, to strengthen the defense ability against toxic materials. For aerobic bacteria, these activities required the consumption of a large amount of ATP provided by respiration with oxygen as the electron acceptor; therefore, oxygen played an important role in maintaining the biological activity in toxic substances. Aniline is limited to 1.0 mg·L$^{-1}$ in the latest amended standards for pollutant

discharge of the textile industry in China. Therefore, biological reinforcement and aeration regulation were necessary measures to make the effluent reach the standard.

### 3.2. Removal Performances of Nitrogen under Different Aeration Rates

Figure 2 illustrates the variation rules of the nitrogen metabolism of the system with the aeration rate. As mentioned before, it would release $NH_4^+$-N into water gradually during the aniline biodegradation. In phase I, R1 had a higher aniline removal efficiency than R0, which would lead to more generated $NH_4^+$-N in R1. However, the average $NH_4^+$-N concentrations in the effluent of R1 and R0 were at the same level (Figure 2a), indicating that the removal effect of R1 on $NH_4^+$-N was better. More than that, it was obvious that R1 always had a better performance of nitrogen removal than R0 as shown in Figure 2d. This possibly was due to the addition of the aniline-degrading strain AD4 in R1, which significantly affected the system performance of nitrogen removal.

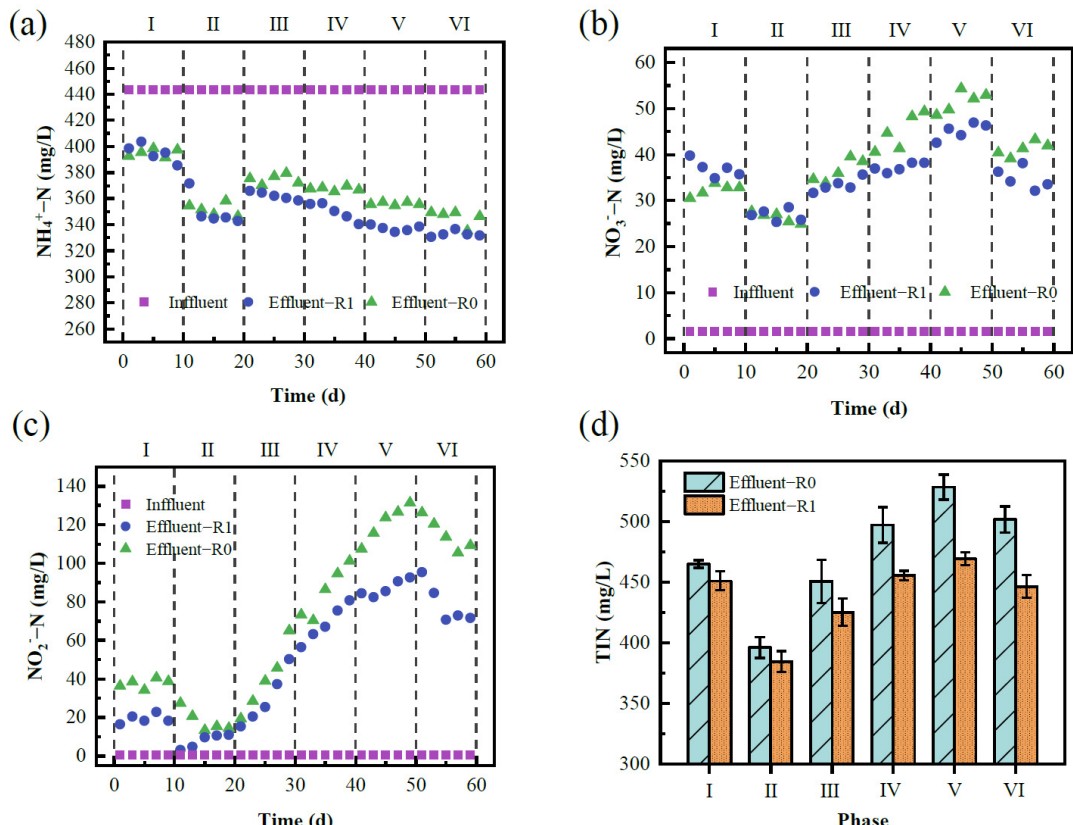

**Figure 2.** Removal performances of reactors for $NH_4^+$-N (**a**), $NO_3^-$-N (**b**), $NO_2^-$-N (**c**) and TIN (**d**) at each phase under different aeration rates.

In terms of the influence of aeration rates on R1 and R0, it could be concluded that a high value of aeration rates, which was related to the value of DO (Table 1), stimulated the degradation of $NH_4^+$-N but caused the accumulation of $NO_3^-$-N and $NO_2^-$-N both in R1 and R0. The TIN concentrations in R1 and R0 were also affected by aeration rates and presented an upward trend. The optimal aeration condition should be 400 mL·min$^{-1}$ in phase II for R1 and R0, because the average TIN concentrations of R1 and R0 (384.69 ± 8.58 and 396.17 ± 8.54 mg·L$^{-1}$, respectively) were lowest under this condition. It was reported that a proper aeration rate was conducive to simultaneous nitrification and denitrification [25]. Thus, $NH_4^+$-N, $NO_3^-$-N and $NO_2^-$-N could be effectively removed in phase II. With the increasing aeration rates, this suitable aeration condition was destroyed, and the system performances of nitrogen removal were inhibited by the impact of the aeration rates. In addition, it should be noted that the TIN concentrations were reduced suddenly when the

aeration rate was increased from 700 mL·min$^{-1}$ to 800 mL·min$^{-1}$. The main reason was due to the reduction of $NO_3^-$-N and $NO_2^-$-N in the effluent (Figure 2b,c). Some kinds of microbes with the ability of aerobic denitrification could remove $NO_3^-$-N and $NO_2^-$-N in an aerobic environment [26]. High aeration rates provided suitable conditions for this kind of microorganism and possibly caused this phenomenon.

**Table 1.** Parameter setting of the reactors.

| | Bioaugmentation System (R1) | | | | | | Control System (R0) | | | | | |
|---|---|---|---|---|---|---|---|---|---|---|---|---|
| Aeration rate (mL air·min$^{-1}$) | 300 | 400 | 500 | 600 | 700 | 800 | 300 | 400 | 500 | 600 | 700 | 800 |
| DO (mg·L$^{-1}$) | 0.91 | 2.21 | 3.65 | 4.22 | 5.65 | 6.78 | 0.92 | 2.14 | 3.52 | 4.62 | 5.73 | 6.81 |
| pH | 7.66 | 7.72 | 7.74 | 7.78 | 7.85 | 7.92 | 7.26 | 7.67 | 7.75 | 7.81 | 7.82 | 7.91 |

In brief, R1 always had a better performance of nitrogen removal than R0. Too high aeration rates were detrimental to the system performance of nitrogen removal. The optimal aeration condition for nitrogen removal was 400 mL·min$^{-1}$ both in R1 and R0. It should be noted that R1 could achieve a satisfactory aniline degradation efficiency, while R0 could not degrade aniline effectively under this aeration condition, reflecting the advantages of bioaugmentation in R1. This was based on the experimental conditions set in this study to meet the physiological requirements of AD4.

*3.3. Microbial Community Characterization*

3.3.1. Alpha Diversity of the Microbial Community

Table 2 and Figure 3b showed the variations in richness and diversity indices in the microbial community. The coverage of all the samples was over 99.5%, indicating the constructed sequence libraries in this study were reliable and enough to represent the microbial diversity. The value of Sobs, representing the number of OTUs in microbial communities, showed a downward trend both in R1 and R0 samples with the increase in aeration rates. This reflected the selection and elimination of species by aniline environment, and it can be speculated that the remaining species constituted the core flora to resist the impact of aniline. Meanwhile, this may also suggest that the increase in aeration rates was partly responsible for the loss of organisms. The indices of Shannon and Simpson were often used to describe the biodiversity. The larger the value of the Shannon index or the smaller the value of the Simpson index, the greater the biodiversity was in a system. It was obvious that under different aeration rates, the index of Shannon was always larger in R1, and the index of Simpson was always larger in R0, showing a higher bacterial richness and microbial diversity in R1. This once again revealed that the application of AD4 had a positive regulatory effect on the system environment and ensured the survival of more species. Particularly, both R1 and R0 under the aeration condition of 400 mL·min$^{-1}$ had the highest alpha diversity; this may suggest that higher nitrogen metabolic activity was beneficial to maintaining microbial diversity.

**Table 2.** Microbial community richness and diversity indices.

| Samples | Sobs | Shannon | Simpson | Ace | Chao | Coverage |
|---|---|---|---|---|---|---|
| C3 | 694 | 3.320 | 0.217 | 791.068 | 775.752 | 0.996 |
| C4 | 604 | 4.173 | 0.040 | 705.887 | 681.410 | 0.996 |
| C7 | 271 | 1.825 | 0.475 | 354.875 | 368.533 | 0.998 |
| C8 | 241 | 1.244 | 0.624 | 308.927 | 306.029 | 0.998 |
| S3 | 532 | 3.635 | 0.077 | 684.786 | 665.235 | 0.995 |
| S4 | 729 | 4.671 | 0.028 | 818.871 | 806.159 | 0.995 |
| S7 | 331 | 2.485 | 0.317 | 396.831 | 396.878 | 0.997 |
| S8 | 228 | 1.720 | 0.497 | 290.520 | 293.808 | 0.998 |

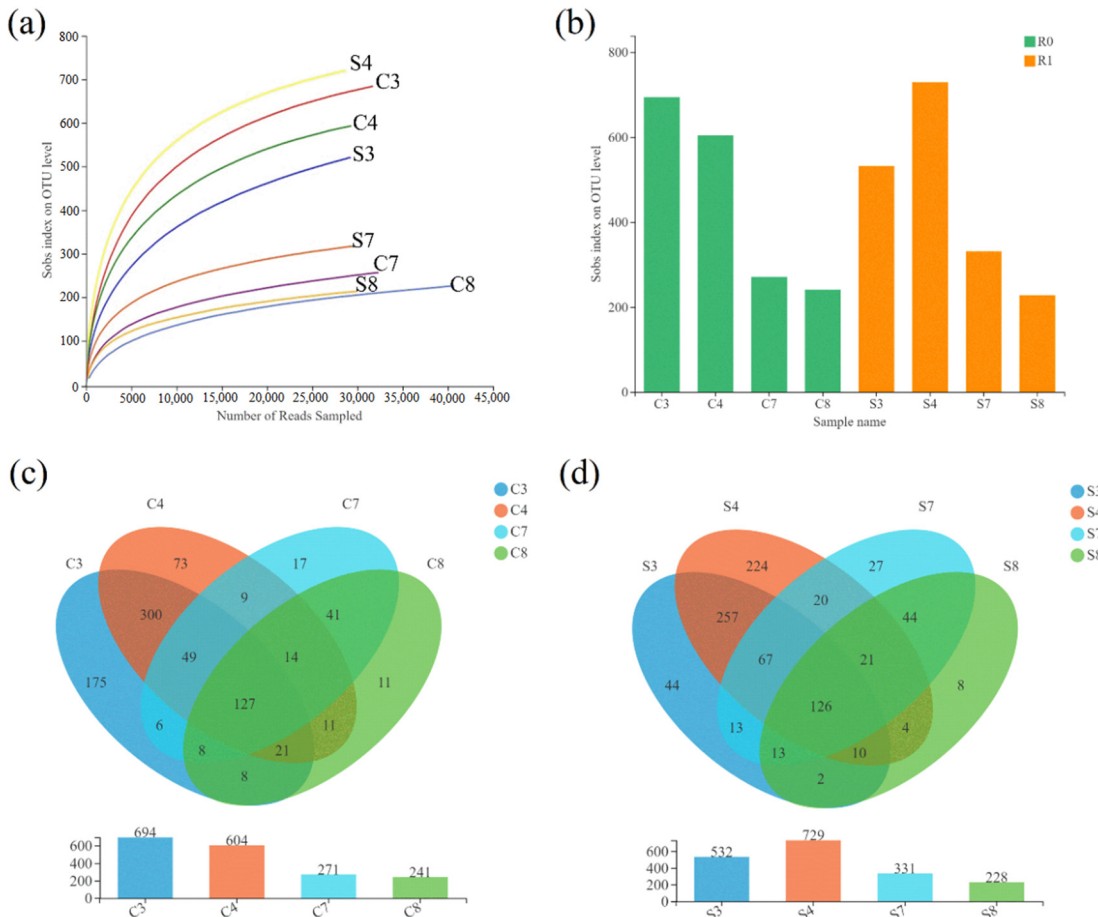

**Figure 3.** Microbial diversities of the eight samples. Rarefaction curves (**a**); alpha diversity estimators (**b**); Venn diagram of R0 (**c**); Venn diagram of R1 (**d**).

Rarefaction curves of the eight samples with a tendency to the plateau are shown in Figure 3a, which indicated that the sequencing depth was adequate in this work. The curves also revealed that the alpha diversity of the microbial community was lower after increasing the aeration rates. As shown in Figure 3c,d, it was obvious that the similarity and overlap of OTUs in sludge samples C3 and C4 were very different from those in sludge samples S3 and S4. Sample S4 contained 224 individual OTUs, which was much higher than sample C4. Furthermore, the number of OTUs decreased from 694 in C3 to 604 in C4. However, the number of OTUs increased from 532 in S3 to 729 in S4. It indicated that adding aniline-degrading bacteria AD4 (*Delftia* sp.) was beneficial for microorganisms in the bioaugmentation system R1 to adapt to the environment in phase II. However, it could also be seen from Figure 3c,d that high aeration rates had an impact both on R1 and R0 and led to a downtrend in the number of OTUs.

### 3.3.2. Microbial Community Structure at Phylum and Class Levels

The relative abundance of the microbial community in the eight samples collected at the end of phase I (C3, S3), II (C4, S4), V (C7, S7) and VI (C8, S8) were shown in Figure 4 on the phylum and class levels. As shown by Figure 4a of the phylum level, the dominant phyla were *Proteobacteria* (21.90–86.76%), *Bacteroidetes* (8.27–60.95%), *Actinobacteria* (1.68–8.57%), *Chloroflexi* (0.91–13.27%) and *Deinococcus-Thermus* (0.01–10.77%), followed by *Acidobacteria* (0.03–1.09%), *Planctomycetes* (0.02–1.57%) and *Patescibacteria* (0.02–3.31%). Among them, *Proteobacteria* and *Bacteroidetes* were the top two abundant phyla in all phases at different aeration rates. *Proteobacteria* belonged to Gram-negative heterotrophic bacteria and could participate in the degradation of multiple pollutants such as nitrogen and aro-

matic compounds [27,28]. The relative abundance of *Proteobacteria* increased a lot both in R0 (from 21.9% to 86.76%) and R1 (from 32.25% to 78.48%) with the increase in aeration rates. It was also reported that *Proteobacteria* could also produce a lot of extracellular polymeric substance, which was beneficial for the cohesion of floc sludge to resist the unfavorable external environment [29].

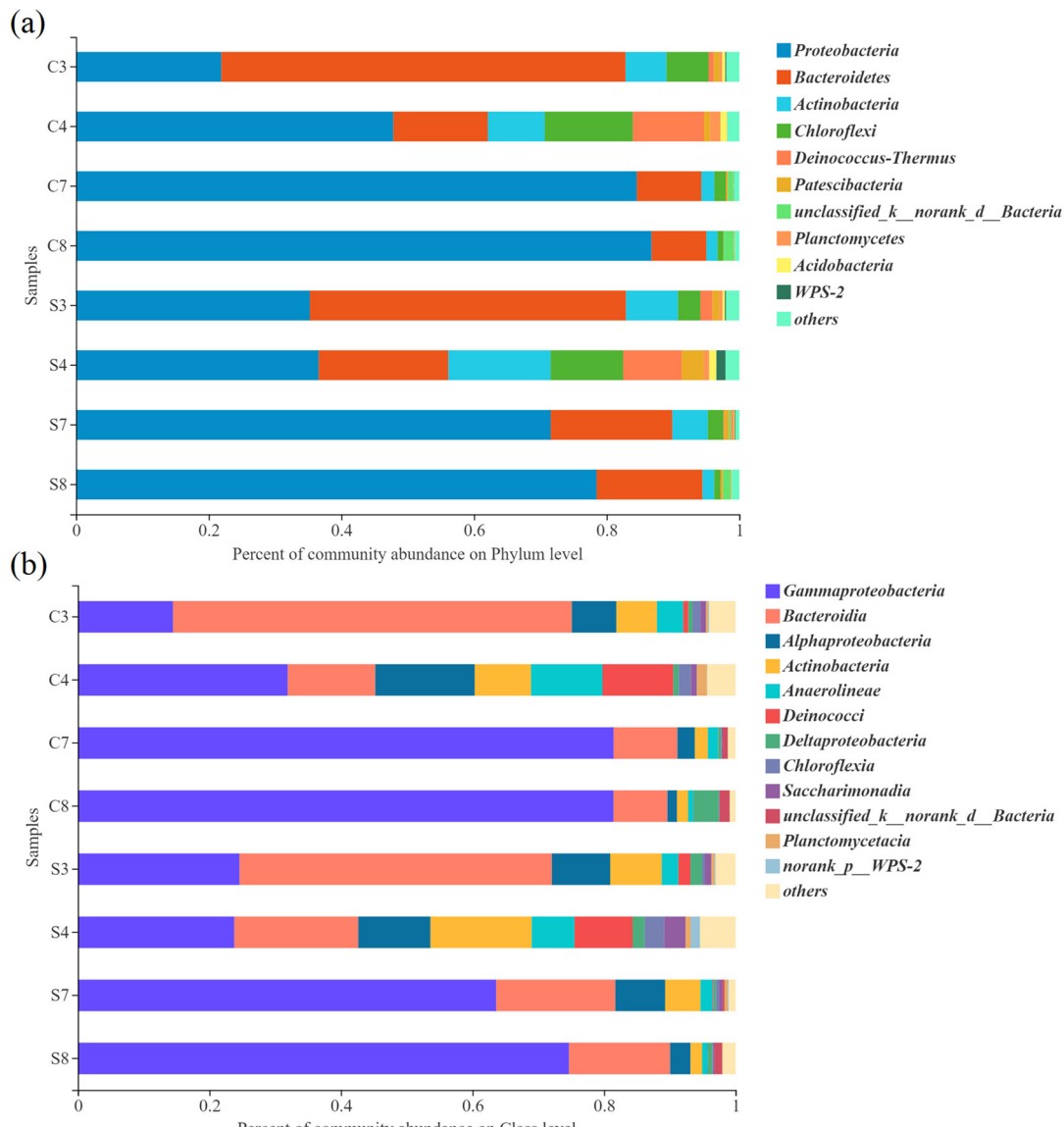

**Figure 4.** Microbial community structures of the eight samples at the phylum level (**a**) and class level (**b**).

The enrichment of phylum *Proteobacteria* suggested that it could adapt to the aniline environment and the impact of high aeration rates. Comparing with *Proteobacteria*, the abundance of *Bacteroides* decreased substantially in R0 (from 60.95% to 8.27%) and R1 (from 47.66% to 15.96%). Although *Bacteroidetes* was reported to contribute to aniline removal [30] and nitrogen metabolism [31], it lost out to *Proteobacteria* during the operation. This may be since the carbon and nitrogen uptake capacity of *Bacteroidetes* was weaker than that of *Proteobacteria* in the toxic environment, and that excessive aeration rates might also adversely affect *Bacteroidetes* [32]. The phylum *Actinobacteria*, which was usually found to participate in the denitrification process [33], had a higher abundance in phase II (increased from 6.15% to 8.57% in R0 and from 7.83% to 15.50% in R1). The higher abundance of *Actinobacteria* in phase II could contribute to the sudden decrease of $NO_3^--N$, and $NO_2^--N$

in the effluent of R1 and R0 from phase I to phase II. Others, such as *Chloroflexi*, *Deinococcus-Thermus* and *Acidobacteria* were also ubiquitous in the biosystem of treating aromatic compounds' wastewater [34,35]. Particularly, the distribution uniformity of phylum in the two systems decreased with the operation, and this again reflected the selectivity of the aniline environment in the microbial community structure. Meanwhile, the uniformity of R1 was always higher than that of R0, which was more conducive to the internal stability of the system.

The compositions of the microbial community at the class level in Figure 4b showed more differences in the eight sludge samples. The five predominant classes in all sludge samples were *Gammaproteobacteria* (14.47–81.46%), *Bacteroidia* (8.20–60.63%), *Alphaproteobacteria* (1.46–15.10%), *Actinobacteria* (1.68–15.40%) and *Anaerolineae* (0.81–10.87%). With the increase in aeration rates, the relative abundance of class *Gammaproteobacteria* increased significantly and turned into the dominant class both in R0 (81.44%) and R1 (74.68%) at the aeration rate of 800 mL·min$^{-1}$. The class *Gammaproteobacteria* belonged to phylum *Proteobacteria* and was a kind of microbe involved in the aniline degradation and denitrification process [36]. The significant enrichment of this kind of microbe during the operation indicated its important role in the aniline-degrading system. Previous research showed that *Bacteroidia* could denitrify [37]. Though *Bacteroidia* was dominant in phase I-II, its relative abundance shifted to 8.20% in R0 and 15.37% in R1 in phase VI. This indicated it could not adapt to the environment with high aeration rates well. By comparing the abundance of *Bacteroidia* between R0 and R1 (13.31% in C4, 18.86% in S4, 9.7% in C7, 18.13% in S7, 8.20% in C8 and 15.37% in S8), it was apparent that this class was highly abundant in bioaugmentation system R1. This result was also positively related to the better $NO_3^-$-N, and $NO_2^-$-N removal performance in R1. There were also enrichments of *Actinobacteria*, *Alphaproteobacteria*, *Anaerolineae*, *Deinococci*, *Deltaproteobacteria*, *Saccharimonadia*, *Planctomycetacia* and *Chlorofloxia* in phase II (8.57%, 15.10%, 10.87%, 10.77%, 0.82%, 0.90%, 1.53% and 1.87% in R0, respectively; 15.40%, 10.97%, 6.50%, 8.84%, 1.84%, 3.24%, 0.79% and 2.97% in R1, respectively). The above microbes were always reported to be dominant in the treatments of aniline wastewater [17,34,36,38], which were responsible for aniline degradation and nitrogen removal.

### 3.3.3. Microbial Community Structure at Genus Levels

The top 50 genera were selected for further analysis on the succession of the microbial community structure at the genus level in Figure 5. At the initial phase I, genera such as *norank-f-Saprospiraceae* (R0—45.83%, R1—19.85%), *Flavihumibacter* (R0—6.57%, R1—10.74%) and *unclassified-f-Burkholderiaceae* (R0—6.15%, R1—13.55%) were dominant genera in both systems. Adding the extra aniline-degrading strain AD4 into R1 shifted the microbial community structure substantially at this phase. Genera such as *Terrimonas* (1.15%), *Acidovorax* (1.34%), *OLB13* (1.01%), *norank-f-Caldilineaceae* (1.07%) and *norank-f-JG30-KF-CM45* (1.03%) were dominant in R0. However, it was found that *Niabella* (1.35%), *Leifsonia* (1.26%), *Cryobacterium* (3.05%), *Chryseolinea* (4.98%), *Truepera* (1.83%), *Unclassified-f-Rhizobiaceae* (3.26%) and *norank-f-NS9-marie-group* (2.22%) were dominant in R1. The AD4 strain might alter the relationship in microbial competition and cooperation and make a difference in the microbial community in R1 [39]. The principal component analysis (PCA) shown in Figure S1 further intuitively proved this. During phase II, the aeration rate was increased further and caused the fluctuation of the relative abundance of some genera.

The genera such as *Truepera*, *Chryseolinea*, *Thermomonas*, *Leucobacter*, *Diaphorobacter*, *Candidatus-Paracaedibacter* and *Pseudaminobacter* showed fluctuation with the increase in aeration rates, but achieved higher abundance at phase II (10.76%, 3.31%, 6.76%, 1.42%, <1%, <1%, 1.97% in R0, respectively, and 8.83%, <1%, <1%, 4.24%, 3.9%, 1.36%, <1% in R1, respectively). This was also positively related to the highest alpha diversity in phase II. The microbial community in the C7, S7, C8 and S8 sludge samples experienced great succession at the genus level. Apparently, genera *Acidovorax* increased significantly and accounted for 69.05%, 55.34%, 78.84% and 70.12% in C7, S7, C8 and S8, respectively. Furthermore,

*Ferruginibacter* also showed much higher abundances when it was exposed to high aeration rates. However, the relative abundance of genera *norank-f-Saprospiraceae, Comamonas* and *Flavihumibacter* appeared as downward trends with the elevated aeration rates, indicating they could not accommodate to the rapidly changing conditions well.

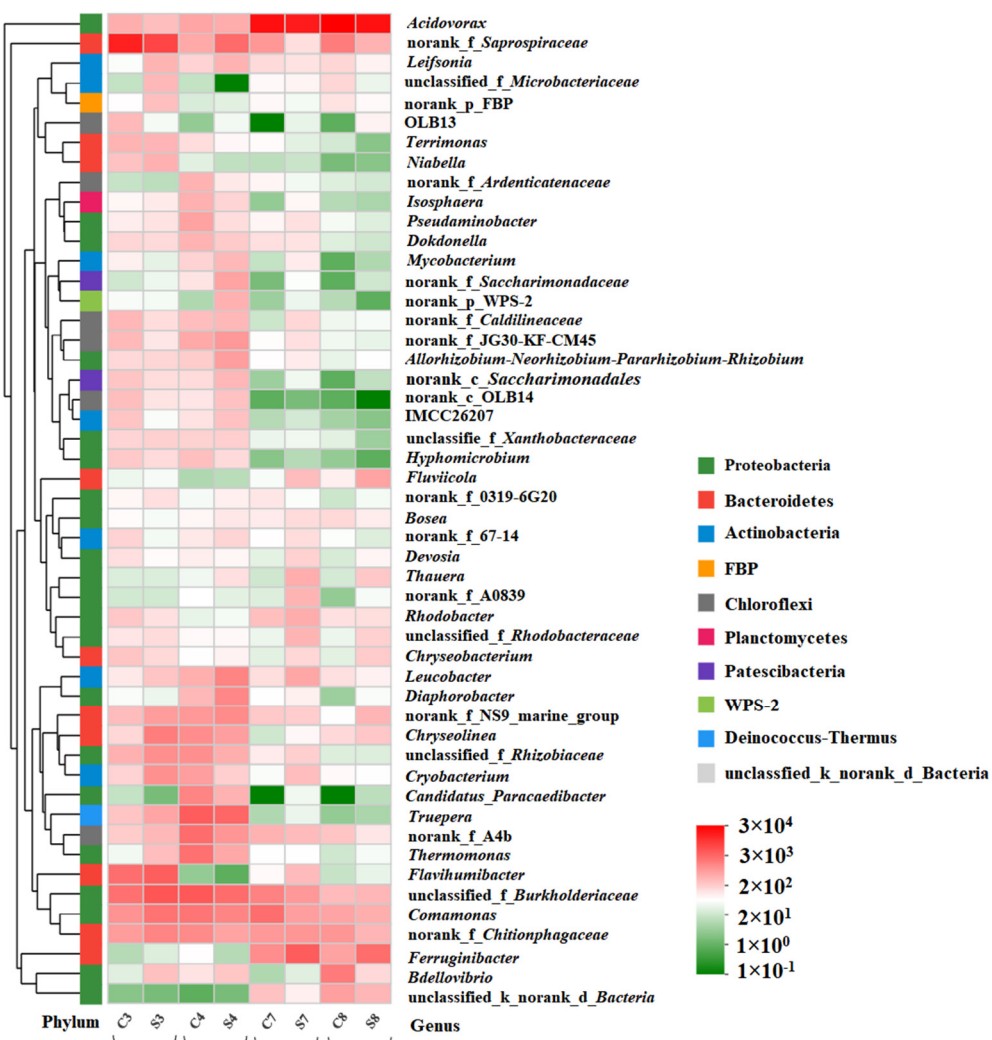

**Figure 5.** Microbial community structures of the eight samples at the genus level.

### 3.4. Variation of Key Functional Groups

To further analyze the connection between microbial community succession and the variation of reactor performances in R0 and R1, the phylogenetic classification of the dominant functional groups at the genus level was performed in Figure 6. Some functional groups in charge of aniline removal such as *Pseudaminobacter, Thermomonas* [40] and *Leucobacter* [41] maintained a higher abundance at 400 mL·min$^{-1}$ aeration rates. This suggested that this aeration condition was favorable to these genera. At 800 mL·min$^{-1}$ aeration, the relative abundance of these genera and other aniline-degrading bacteria such as *Comamonas* [40] and *Niabella* [42] decreased to a low level. By contrast, the genus *Acidovorax*, which could use aniline as the sole source of carbon and energy [40], accumulated significantly both in R0 and R1 at high aeration rates (Figure 6c). It could be concluded that aeration rates shifted the structure of functional microorganisms in aniline degradation significantly. The biodiversity decreased under the impact of high aeration rates, which was not good for the system steady. Additionally, the predominant genus *Acidovorax* might take up too much of the carbon source and further inhibit other aniline-degrading bacteria. There was an enrichment of genus *Delftia* which was highlighted in red in Figure 6b, includ-

ing the strain AD4 (*Delftia sp*.). The shift of aeration rates improved its abundance, proving its colonization in this system. However, the abundance of *Delftia* was not at a high level and might be possibly inhibited by genus *Acidovorax* too. On the other hand, the genus *Delftia* might alleviate the competition between genus *Acidovorax* and other functional groups, leading to an *Acidovorax* abundance ratio of R1 that was less than R0, which was beneficial for the biodiversity in the system.

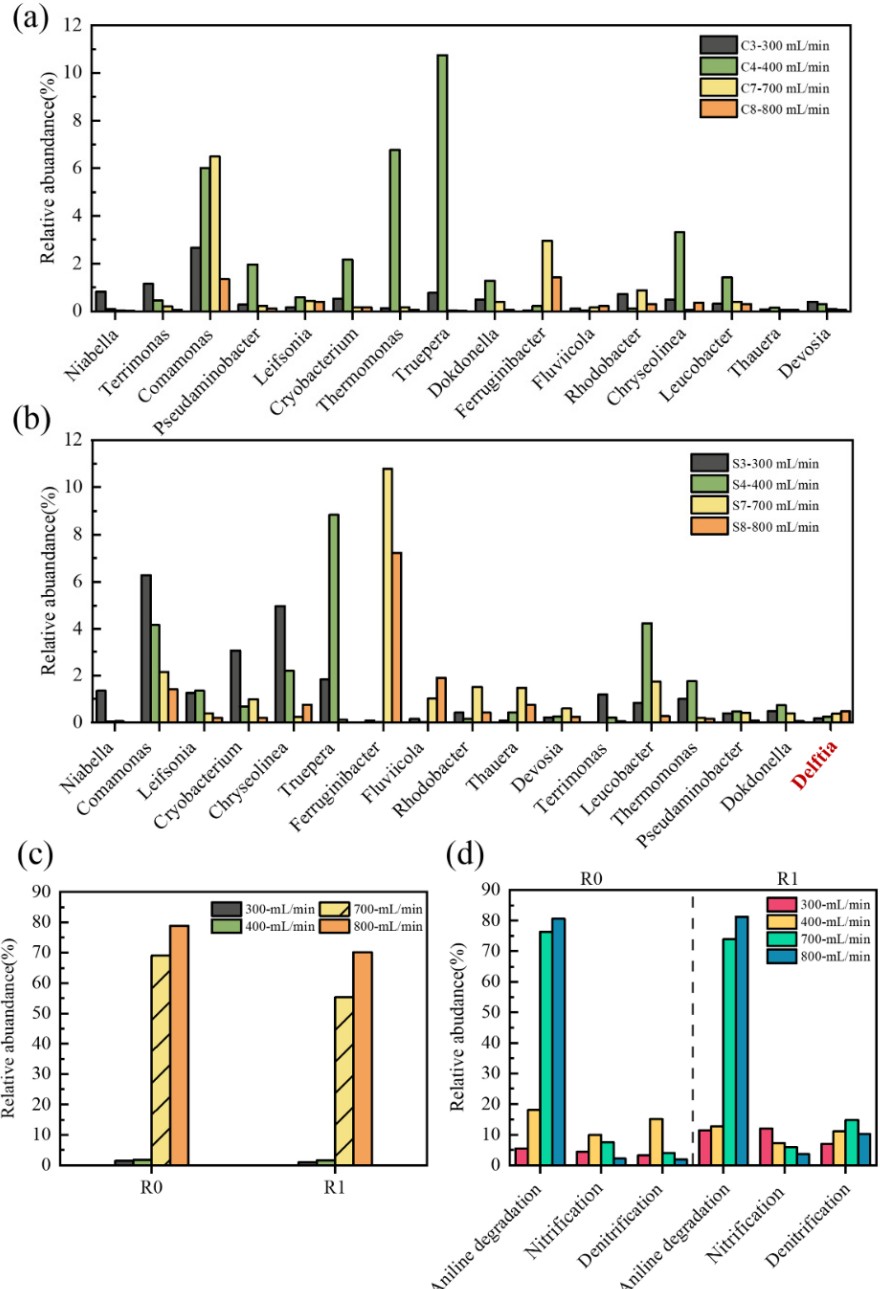

**Figure 6.** The distribution of functional microfloras. Phylogenetic classification of key functional groups in R0 (**a**) and R1 (**b**); the relative abundance of *Acidovorax* in R0 and R1 (**c**); the relative abundance of key functional groups in R0 and R1 (**d**).

The relative abundances of ammonia-oxidizing bacteria (AOB) and nitrite-oxidizing bacteria (NOB) were at a low level in this work (<0.5% at 300, 400 mL·min$^{-1}$ aeration rates and not detected at 700, 800 mL·min$^{-1}$ aeration rates). Meanwhile, the nitrifier *Devosia* [43] possessed less than 1% abundance, although it existed in both R0 and R1. On the one hand, these bacteria were slow growing due to long generation cycles. On the other hand, a high

concentration of aniline usually inhibited the growth and activity of these bacteria [44]. Despite this, *Chryseolinea* and *Rhodobacter* could carry out heterotrophic nitrification-aerobic denitrification (HN-AN) [17]. Thus, these genera might contribute together to the removal of $NH_4^+$-N in this work.

In terms of the denitrifiers, genera *Truepera*, *Cryobacterium* and *Dokdonella* have been reported to be capable of denitrification [45–47]. The abundances of these genera reached a relatively high value at the 400 mL·min$^{-1}$ aeration rate. However, there was an apparent decline in their abundances under 700 and 800 mL·min$^{-1}$ aeration rates. Hence, this might be responsible for the accumulation of $NO_3^-$-N and $NO_2^-$-N with increasing aeration rates. This result also suggested that the 400 mL·min$^{-1}$ aeration rate was more favorable for these denitrifiers. Notably, three kinds of denitrifiers, *Fluviicola* [48], *Thauera* and *Ferruginibacter* were detected at 700, 800 mL·min$^{-1}$ aeration rates. *Thauera* has been confirmed with the ability of denitrification in both aerobic and anoxic environments [49]. In addition, genus *Ferruginibacter* was reported to play an important role in denitrification and prefer an aerobic environment [50,51]. Thus, they could adapt to this environment more easily. These above microorganisms, especially genus *Ferruginibacter*, had much higher abundances in R1 than in R0 (2.96% in C7, 1.42% in C8, 10.78% in S7 and 7.23% in S8). Furthermore, the abundance of denitrification groups was also much higher in R1. This might be the main reason why R1 had a better performance of denitrification under high aeration rates.

Based on these results, it was demonstrated that the 400 mL·min$^{-1}$ aeration rate could keep higher biodiversity of functional genera and was more favorable for the coexistence of these functional microbes. In addition, the bioaugmentation system R1 with an enrichment of genus *Delftia* might alleviate the toxicity of aniline to create a more suitable environment for other functional groups, or it might change the competition and cooperation relationships in the microbial community. As a result, there were some kinds of functional genera with higher abundances detected in R1 than in R0.

## 4. Conclusions

In the present research, the influence of the aeration rate on the bioaugmentation-SBR system R1 and control system R0 was investigated systematically. The purpose was to provide guidance for practical engineering to solve the problem of aniline wastewater degradation, and to realize the low cost but high efficiency of the process operation. The increase in aeration rates could significantly improve the aniline degradation efficiency, and the bioaugmentation system R1 could achieve almost 100% removal efficiency. Meanwhile, aeration rates also affected the nitrogen metabolism of the systems. The increase in aeration rates inhibited microbial diversity to a certain extent and significantly changed the structure of the microbial community. Under the regulation of high efficiency aniline-degrading bacteria (AD4), species diversity and species distribution uniformity in the enhanced reactor were higher than those in the control group. However, there are still some defects in this study that need further exploration in the future. The control of biological loss is of great significance to the biofortification system, so it is necessary to optimize the adding mode of strains. Meanwhile, future studies should consider aniline as the sole source of carbon and nitrogen, and the response rules of functional genes and metabolic pathways should be included in the discussion.

**Supplementary Materials:** The following supporting information can be downloaded at: https://www.mdpi.com/article/10.3390/w14244096/s1, Figure S1: Principal component analysis between bacterial communities in different samples under different aeration rates.

**Author Contributions:** Conceptualization, J.S., C.W. and Q.Z.; methodology, C.W. and H.P.; validation, Y.L. and H.W. (Hua Wei); investigation, J.S. and H.P.; data curation, J.S.; writing—original draft preparation, J.S.; writing—review and editing, C.W., H.P., Q.Z., Y.L., H.W. (Hua Wei) and H.W. (Hongyu Wang); supervision, J.S.; funding acquisition, J.S. and Q.Z. All authors have read and agreed to the published version of the manuscript.

**Funding:** This work was financially supported by the Henan Key Laboratory of Industrial Microbial Resources and Fermentation Technology, Nanyang Institute of Technology (HIMFT20210205 and HIMFT20200205); the Scientific and Technological Projects of Henan Province (212102310278); the Scientific and Technological Projects of Nanyang City (KJGG056); the Doctoral Research Start-up Fund Project of Nanyang Institute of Technology (510161); and the Interdisciplinary Sciences Project, Nanyang Institute of Technology.

**Institutional Review Board Statement:** Not applicable.

**Informed Consent Statement:** Not applicable.

**Data Availability Statement:** The data presented in this study are available on request from the corresponding author.

**Conflicts of Interest:** The authors declare no conflict of interest.

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
