# Peer review of "Optimization of Aeration Rate—Low Cost but High Efficiency Operation of Aniline-Degrading Bioaugmentation Reactor"

_water, doi:10.3390/w14244096_

Round 1
Reviewer 1 Report
The manuscript “Effect of aeration rate on the treatment of high concentration aniline wastewater by bioaugmentation system” describes an interesting study that tested the effect of aeration rates on the degradation of aniline in two sequencing batches reactors with and without the bacterial fluid of the AD4 strain, bioaugmentation system (R1) and control system (R0), respectively. The study is interesting however, some minor corrections need to be done. For more details please check the attached pdf file.

Reviewer 2 Report
The manuscript “Effect of aeration rate on treatment of high concentration aniline wastewater by bioaugmentation system” set up a bioaugmentation system (R1) by adding highly efficient aniline-degrading strain AD4 (Delftia sp.), and a control system (R0) without the addition of AD4. Then, they investigated the effect of aeration rates on aniline and TIN removal in two systems, and suggested that 400 mL/min was the optimum aeration rate for both bioaugmentation system and control system. However, the reason why 400 mL/min was the optimum aeration rate was not adequately discussed. More discussion is suggested about the influence of adding the bioaugmentation strain AD4 on microbial community and the variation of function bacteria. Besides, the effect of aeration rates on the variation of AD4 relative abundance also could be discussed further.
1. Page 3 line 125-127: From Fig.1, the removal efficiency of aniline in R1 was higher than R0. However, as you mentioned before, AD4 was a kind of efficient aniline degrading bacteria. After adding 5% volume fraction of AD4 bacterial fluid into R1, it strengthened the aniline removal efficiency significantly, but maybe you should give more discussion to prove this enhancement effect was a bioaugmentation effect rather than the degradation by AD4 alone. Besides, more explanations on “(AD4) reducing the toxicity inhibition effect of aniline on activated sludge” should be given.
2. Page 4 line 132: In Fig. 1 (b) and (c), please clarify whether the concentration of aniline here was the concentration of the end of each phase or the average concentration of each phase (10 d).
3. Page 4 line 136: Maybe give detailed values (for example, from …% to …%) would be better.
4. Page 4 line 145-149: What role did oxygen supply play here? please give more explanations or delete “In the toxicity environment, … against toxic materials.”.
5. Page 5 line 162: Why did you choose such high NH4+-N concentration in the synthetic wastewater? What’s your reason for choosing the synthetic wastewater with these compositions?
6. Page 5 line 162: Could you please provide the data of the variation of TN?
7. Page 6 line 176-183: In Table 1, DO concentration was 3~6 mg/L in R1 and R0, it also belonged to the aerobic condition. Thus, could you please give more discussion on why TIN removal effect is destroyed as the DO concentration increasing, and why it suddenly gets better in phase VI?
8. Page 6 line 192: Please provide the project number (NCBI Sequence Read Archive) of your microbial community analysis.
9. Page 7 line 212: Fig. 3, make sure the figure with high resolution.
10. Page 7 line 226: The strain AD4 belonged to Delftia sp., please discuss its abundance variation on genus level? How about the persistent bioaugmentation effect of AD4 during the long-term operation period.
11. Page 9 line 248-251: provide the reference here?
12. Page 10 line 298: give more discussion on PCA here.
13. Page 13 line 370: In your work, the reasons why AD4 induced a bioaugmentation effect were most based on speculation, please provide more experiment data to discuss the bioaugmentation effect of AD4.
Reviewer 3 Report
Reviewer Report
Song et al. Exploring the effects of varying aeration rates on the bioaugmentation and control systems treating a high concentration of aniline wastewater. It seems to be an important study and can be considered for publication in this Journal. However, the following suggestions are recommended:
i. In the abstract, the author can state the perspective of this study for readers.
ii. In the last paragraph of the introduction, the Author needs to clearly state the novelty of this paper together with detailed aims and future prospects of this study. In particular, there are several reviews in the literature on bioaugmentation systems. The author needs to explain these together in this study.
iii. In conclusion, author need to articulate the limitations, prospect and future direction of research in the current topics to make the study stand alone.
iv. Authors need to follow the journal format fully in case of Reference list. For example, Journal Abbreviation, heading and subheadings etc.
v. All Table font and legend should be uniform and clear to understand.
vi. Some Figure captions are not clear. So please make these bigger and clear.
vii. There are also several typos and grammatical mistakes throughout the manuscript. Author needs to revise these carefully.
viii. Author needs to summarize the major findings of this study in a figure/scheme/graphical abstract. This could be interesting part of this study.
ix. Author can compare the advantages or disadvantages of bioaugmentation systems if possible.
x. Please provide some practical applications/benefits of this study.

Reviewer 4 Report
The manuscript studies the role of aeration rate on the treatment of aniline wastewater through SBR. The technical English and formatting of the manuscript needs improvement, and more discussion needs to be added in the manuscript. Specific comments on the manuscript can be found in the attached annotated pdf.

Reviewer 5 Report
This research designed the bioaugmentation system to treat high concentration aniline wastewater and the influence of aeration rate was also studied, which play a great role for the application of bioaugmentation system for environmental remediation. This work provided sufficient data and high quality Figures to illustrated the conclusions. However, the sufficiency and innovation of this bioaugmentation system should be provided in the Introduction. Additionally, an interesting Title should be provided to attract more readers. After revision, this submitted manuscript can be published at “Water”.
Round 2
Reviewer 2 Report
Authors revised as each comment and might be accepted without further corrections.
Reviewer 3 Report
The authors have improved the manuscript to a notable extent in this revised version. I, therefore, recommend its publication in this Journal at this stage.
Reviewer 4 Report
None